# Biodegradation and Non-Enzymatic Hydrolysis of Poly(Lactic-*co*-Glycolic Acid) (PLGA12/88 and PLGA6/94)

**DOI:** 10.3390/polym14010015

**Published:** 2021-12-21

**Authors:** Yue Wang, Maria A. Murcia Valderrama, Robert-Jan van Putten, Charlie J. E. Davey, Albert Tietema, John R. Parsons, Bing Wang, Gert-Jan M. Gruter

**Affiliations:** 1Van ‘t Hoff Institute for Molecular Sciences (HIMS), Faculty of Science, University of Amsterdam, Science Park 904, 1098 XH Amsterdam, The Netherlands; y.wang6@uva.nl (Y.W.); m.a.murciavalderrama@uva.nl (M.A.M.V.); robert-jan.vanputten@avantium.com (R.-J.v.P.); c.j.e.davey@uva.nl (C.J.E.D.); 2Institute for Biodiversity and Ecosystem Dynamics (IBED), Faculty of Science, University of Amsterdam, Science Park 904, 1098 XH Amsterdam, The Netherlands; a.tietema@uva.nl (A.T.); j.r.parsons@uva.nl (J.R.P.); 3Avantium Support BV, Zekeringstraat 29, 1014 BV Amsterdam, The Netherlands; bing.wang@avantium.com

**Keywords:** bio-based plastic, biodegradation, plastic, poly(lactic-*co*-glycolic acid), polyester, high-throughput, respirometer, paper coating, packaging

## Abstract

The predicted growth in plastic demand and the targets for global CO_2_ emission reductions require a transition to replace fossil-based feedstock for polymers and a transition to close- loop recyclable, and in some cases to, biodegradable polymers. The global crisis in terms of plastic littering will furthermore force a transition towards materials that will not linger in nature but will degrade over time in case they inadvertently end up in nature. Efficient systems for studying polymer (bio)degradation are therefore required. In this research, the Respicond parallel respirometer was applied to polyester degradation studies. Two poly(lactic-*co*-glycolic acid) copolyesters (PLGA12/88 and PLGA6/94) were tested and shown to mineralise faster than cellulose over 53 days at 25 °C in soil: 37% biodegradation for PLGA12/88, 53% for PLGA6/94, and 30% for cellulose. The corresponding monomers mineralised much faster than the polymers. The methodology presented in this article makes (bio)degradability studies as part of a materials development process economical and, at the same time, time-efficient and of high scientific quality. Additionally, PLGA12/88 and PLGA6/94 were shown to non-enzymatically hydrolyse in water at similar rates, which is relevant for both soil and marine (bio)degradability.

## 1. Introduction

In 2019, 368 million tons of plastic were produced worldwide, of which an estimated 6–17 million tons accumulates in the environment annually [1,2]. Nearly all of these plastics (>99%) are produced from fossil resources, annually consuming about 4–8% of oil production for material feedstock and for their production (roughly half–half) [3]. Plastic production is expected to triple to more than 1 billion tons by 2050, with an associated annual plastic CO_2_ footprint of 2.8 billion tons (2.8 Gt) [4]. This is incompatible with the global CO_2_ emission reduction targets required to minimise global warming and climate change. Therefore, as a world, we will need to transition from plastics produced from fossil resources to plastics produced from carbon already “above the ground”. There are only two alternative feedstocks for producing virgin plastic materials: biomass and CO_2_ (via carbon capture and utilisation (CCU)) [5]. Next to sustainable feedstocks, other options to minimise the footprint of plastics are the well-known three R’s: reduce, reuse and recycle. Apart from the effects of plastics on global warming, plastics also have a waste problem. Plastics typically end up in the environment when they have no residual value, and this is why the so-called plastic soup mainly consists of unrecyclable single-use packaging waste (more than 68% polypropylene (PP) and polyethylene (PE)) [6]. Although biodegradable plastics will often not be a viable solution for this problem, plastics of the future should also have design features that address end-of-life (such as closed-loop recyclability) and fate-in-nature (even slow biodegradation will avoid accumulation over decades or even centuries). Although readily biodegradable plastic is not desired for some applications, the fact that it biodegrades is important, as it will avoid plastics accumulating as is the case for current materials such as polyethylene terephthalate (PET) and polyolefins.

Poly(lactic acid) (PLA) was the most produced bio-based plastic by volume in 2020, with applications in packaging, plastic bags and disposable cutlery [7]. It is commercially produced from lactide, the cyclic diester of lactic acid (LA), which in turn is produced by fermentation of glucose [8]. Similar to LA, glycolic acid (GA) can be made from biomass, but potentially also from CO_2_ [9]. LA/GA copolymers, poly(lactic-*co*-glycolic acid) (PLGA), have been used as materials for the synthesis of absorbable sutures and are being evaluated in the biomedical field [10]. In the European Horizon 2020 project “OCEAN”, our group is involved in the development of a continuous multistep process from CO_2_ to oxalic acid and derivatives such as glycolic acid, starting with the electrochemical reduction of CO_2_ [11]. We are interested in evaluating these polymers for other applications, such as paper coating in packaging. PLGA copolyesters with high lactic acid (LA) content and lack good oxygen and moisture barriers. Therefore, PLGAs with over 50% glycolic acid content have been synthesised and studied in our group [9]. Increased thermal stability was observed with increasing glycolic acid content [9]. The barrier property assessment revealed that increasing glycolic acid content in PLGA copolymers enhances the barrier to both oxygen and water vapour. At room temperature and a relative humidity below 70%, the PLGA copolymers with high glycolic acid content outperform non-oriented PET on barrier properties [9]. This shows the great potential for these copolymers for application in barrier films. The expected short lifetime of these copolymers makes them promising candidates for certain short lifetime applications. Therefore, studying the biodegradability of two representative PLGA copolyesters (PLGA12/88 and PLGA6/94) with desired barrier properties is important for further application. 

The process of plastic biodegradation typically has three main phases: (1) disintegration/fragmentation/deterioration, (2) depolymerisation (hydrolysis for polyesters), and (3) mineralisation, which involves microbial utilisation of monomers and oligomers from the second phase leading to the release of mainly CO_2_ and H_2_O under aerobic conditions (Figure 1) [12,13,14]. Heat, light, mechanical stress, humidity and microorganisms are all drivers that can play a role in the first phase. This initial phase results in modifications in polymer physical (e.g., morphology, weight loss) and mechanical properties (e.g., ductility and tensile strength) and the release of micro- and nanoplastics into the environment. After polymers break down to oligomers and monomers, microorganisms can take them up and utilise them as substrates for metabolism and biomass growth [15]. In the third phase, polymer carbon is converted into CO_2_ (mainly) and biomass under aerobic conditions. 

To assess biodegradation of plastic, mass loss, spectroscopy (such as gel permeation chromatography (GPC), infrared spectroscopy (IR), nuclear magnetic resonance (NMR) spectroscopy), visual analysis (observation, scanning electron microscope (SEM)) and respirometry are the most commonly used methodologies [16]. The first three methods all rely on sample collection and/or extraction of plastics. If it is in soil and sediment, separation and collection of plastic samples is not easy when samples consist of small fragments, for instance due to disintegration. Furthermore, monitoring the mass loss of plastic may not represent biodegradability and lead to false positive results, as it could be the result of the release of microplastic particles (and the loss of volatile and soluble components). Released microplastics may cause other potential threats to the environment and human health [17,18]. Extraction could be a solution but requires considerable effort on method development, especially for novel materials. Conversely, monitoring the process of mineralisation by CO_2_ (and/or CH_4_ under anaerobic conditions) evolution and O_2_ consumption, is not specific to the material [13]. This means that it could be applied to novel materials without too much customisation. Specifically, the conversion of polymer-derived carbon into CO_2_ is the direct demonstration of active polymer biodegradation without accumulation of intermediates [19]. 

Therefore, the analysis and quantification of CO_2_ released from polymer degradation is required for assessing the biodegradability of polymers [19]. 

Standard biodegradation tests typically require triplicates for each test material as well as blanks and references. The fact that polymers are solids makes it difficult to mix them homogeneously with inoculum, which is why more replicates are recommended. Considering the slow biodegradation of most plastics, biodegradation tests are usually time-consuming. Consequently, many parallel reactors are necessary to improve time efficiency (per sample). In our lab, we use a Respicond (A. Nordgren Innovations AB, Djäkneboda 99, SE915 97 Bygdeå, Sweden) 95-vessel parallel testing platform with automated CO_2_ release monitoring [20]. An individual vessel is a closed system with a hydroxide solution inside to trap the evolved CO_2_. The amount of absorbed CO_2_ is measured based on the conductivity change of the hydroxide solution [21]. This setup has been used since 1986 [22]. It has been used to study the effect of nutrients and contamination on soil respiration and for studying the decomposition of plant material [23,24,25,26]. It has been used to monitor CO_2_ accumulation in a study on the decomposition of tobacco roots [23], as well as CO_2_ fluxes (carbon mineralisation rate) of soil to assess the effect of land use on soil organic carbon stocks and sequestration [24]. There are over 100 publications reporting the use of this equipment, but to the best of our knowledge, this paper reports the first application of this equipment for plastic biodegradation studies.

The aim of this research is to study the biodegradability of two representative high barrier PLGA copolyesters in soil (PLGA12/88 and PLGA6/94) by measuring CO_2_ evolution in the Respicond. In order to better understand the underlying degradation pathways and mechanisms, the biodegradation of the monomers (GA and LA) in soil and the non-enzymatic hydrolysis of PLGA in water were also researched. 

## 2. Methods and Materials

### 2.1. Materials

Glycolic acid (99%) and l-(+)-lactic acid (≥88%) were purchased from Aldrich (Darmstadt, Germany) and Fisher Scientific (Leicestershire, UK), respectively and used as received in the monomer biodegradation studies. Tin (II) 2-ethylhexanoate (Sn(Oct)_2_) was acquired from Aldrich. Lactide was acquired from Corbion (Gorinchem, The Netherlands) and was used for the synthesis of PLA by ring opening polymerisation together with 1-dodecanol (98%) from Merck. d(+)-glucose monohydrate and cellulose (powder, 20 µm average particle size) were purchased from Merck (Darmstadt, Germany) and Sigma–Aldrich (Schnelldorf, Germany), respectively. Deuterated water (99.9% D) and dimethyl sulfoxide (DMSO, ≥99.9%) were purchased from Aldrich and Fisher Scientific, respectively. Sodium deuteroxide solution (40 wt.% NaOD in D_2_O, 99.5 atom % D) was purchased from Aldrich.

### 2.2. Soil

The soil used was collected from an active agricultural field in Vredepeel, in Limburg province in the south of the Netherlands (51°32′25.8″ N, 5°51′15.5″ E) and was previously described by Schlemper et al. (2017) [27]. It was sieved through a 4 mm mesh and stored in air-dry conditions. It was dried at 40 °C for 70 h before use. The soil properties as provided (*) or as re-established are listed in Table 1.

### 2.3. Polymer Synthesis and Characterisation

**PLGA12/88** and **PLGA6/94**: Both types of PLGA copolymers were synthesised via direct polycondensation. Initially, the required amounts of lactic acid (4.91 or 2.53 g) and glycolic acid (15.1 or 17.5 g) were weighed in a round bottom flask, fit with a mechanical stirrer (95 rpm), a nitrogen inlet and a nitrogen outlet. 0.02 mol% of (Sn (Oct)_2_) with respect to the total monomer load was used as catalyst. Subsequently, the system was heated in an oil bath at 200 °C for 4.5 h under nitrogen atmosphere (20 mL min^−1^) at ambient pressure with removal of water as the side product. After 4.5 h, the temperature was increased to 210 °C and the pressure was gradually reduced to 12 mbar within 2.5 h. For both copolymers, a light yellow product was recovered.

**PLA:** Ten grams of lactide was weighed in a round bottom flask with 0.01 mol% of (Sn (Oct)_2_) as catalyst and 1-dodecanol as initiator. The system was first submitted to 3 cycles of vacuum and nitrogen (5 min each) at 40 °C. Afterwards, it was heated in an oil bath at 150 °C and the temperature was increased to 195 °C within 8 h. The reaction proceeded under nitrogen atmosphere (20 mL min^−1^) at ambient pressure. Finally, a light transparent product was recovered.

PLGA12/88 exhibited an amorphous structure, unlike their homopolymers PGA and PLA, which are known to be semi-crystalline at these same testing conditions in DSC (10 °C/min). PLGA6/94 showed a semi-crystalline structure with lower T_g_ than PLGA12/88 (31 vs 40 °C) and T_m_ = 186 °C (Table 2).

### 2.4. Soil and Test Material Preparation for Biodegradation

To each of the 250 mL vessels, 15 g of (wet) soil mixture (~12.5 g dry soil), with or without test material, was added. In order to adjust the moisture level of all soil mixtures to about 50% of the field capacity, a mineral salts solution [28] (OECD TG310, Table A1) was added slowly to the dry soil in a soft plastic bag. This plastic bag was massaged in order to homogeneously moisturise the soil. The pH was determined by using the method published by Hendershot et al. using 0.01 M CaCl_2_ [29]. After this moisture adjustment, the pH of the soil was 5.9. 

Apart from five blanks, typically the amount of organic carbon introduced by the test substance to the dry soil was kept at around 5 mg C g^−1^ dry soil. This means the amount of adding test material is equivalent to approximately 62.5 mg C per vessel. The resulting ratio of carbon to nitrogen in the soil containing test material (C:N) was around 12.5:1. The monomer and cyclic diester were dissolved in the mineral salt solution prior to mixing with dry soil. The non-soluble polymer test materials were added as ground powder to the dry soil and subsequently mixed thoroughly by shaking the sealed plastic bags. After even distribution of plastic particles in the fine soil was observed, the moisture was added as described. Lastly, soil with test material was divided over multiple vessels: soluble test materials in triplicate and non-soluble test materials in five replicates. Powdered cellulose was used as a reference material for the polymer biodegradation tests, in triplicate [30]. The pH of the soil mixture was measured before and after incubation.

### 2.5. Soil Biodegradation Testing

The Respicond automated respirometer with 95 vessels was used [20]. The biodegradation tests were performed in the dark, in closed vessels which were maintained at a constant temperature of 25 °C. Figure 2 shows the schematic setup of an individual vessel. CO_2_ evolved from the test medium is trapped by a hydroxide solution inside the vessel (Reaction (1)), which converts the hydroxide into carbonate and thus changes the conductivity of the solution.
(1)$$2\;{\mathrm{OH}}^{-}\;(\mathrm{aq})+{\mathrm{CO}}_{2}\;(g)\;\text{→}\;{\mathrm{CO}}_{3}^{2-}\;(\mathrm{aq})+H_{2}O\;(l)$$

The conductivity of a potassium hydroxide solution (KOH) decreases with increased CO_2_ absorption [21]. The conductance of the KOH solution can be measured and recorded at user-defined intervals (≥20 min): the present study employed 1 h intervals. The amount of absorbed CO_2_ (mg) was calculated (Equation (2)) [31] by the Respicond’s native software.
(2)$${\mathrm{CO}}_{2\text{\_}\mathrm{amount}}=A\frac{C_{t0\;}{-\;C}_{t}}{C_{t0}}$$

A (mg) is a constant dependent on the KOH concentration. C_t0_ (S) represents the conductance of the initial KOH solution and C_t_ (S) represents the conductance of the KOH (+K_2_CO_3_) solution at time t. In this study a 0.6 M KOH (A = 219 mg) solution was used, and in each CO_2_-trap, the solution was refreshed before the maximum CO_2_ absorption was reached.

The relative amount of substrate converted into CO_2_, defined as the degree of biodegradation (Dt, %) of a test material at time t was calculated according to Equation (3): (3)$$\;\mathrm{Dt}=\frac{{\mathrm{CO}}_{2\text{\_}\mathrm{sample}}{\;-\;\mathrm{CO}}_{2\text{\_}\mathrm{blank}}}{{\mathrm{ThCO}}_{2}}\;\times \;100$$

Here CO_2_sample_ (mg) represents the amount of accumulated CO_2_ evolved from a vessel containing soil and test material at time t. CO_2_blank_ (mg) is the average amount of accumulated CO_2_ of the blanks (soil without test material) at time t. ThCO_2_ (mg) is the maximum amount of CO_2_ that could theoretically evolve from the test material, based on the amount added.

### 2.6. Hydrolysis

Polymers were ground into a powder and sieved with a 425 µm screen. About 10 mg polymer was added to 1 mL D_2_O with 2.0 mg mL^−1^ DMSO as a standard in a 5 mm NMR tube (SP Industries, Vineland, NJ, USA). These tubes were subsequently sealed by melting to avoid water evaporation over time and stored at a controlled temperature of 25 °C. The hydrolysis experiments were performed in triplicate over 116 weeks. An Avance III 400 MHz NMR spectrometer (Bruker, Billerica, MA, USA) was used to measure (^1^H NMR) soluble hydrolysis products. Samples were typically measured once per week and in the latter stages once per month. Glycolic acid (GA) and lactic acid (LA) resulting from polymer hydrolysis are soluble in D_2_O and can therefore be quantified (Equation (4)), allowing determination of the degree of hydrolysis (Equation (5)). The complete hydrolysis of copolymers was forced by adding over 100 mg sodium deuteroxide solution to NMR tubes at the end of the hydrolysis experiment. After all solid was dissolved, ^1^H NMR spectra were obtained, and the ratio of monomers (LA/GA) was calculated (Equation (6)).

The amount of dissolved GA or LA was calculated according to Equation (4):(4)$$C_{x}=\frac{I_{x}}{I_{\mathrm{DMSO}}}\;\times \;\frac{N_{\mathrm{DMSO}}}{N_{x}}{\;\times \;C}_{\mathrm{DMSO}}$$

Here, I represents the peak area, N the number of protons corresponding to the integrated peak(s) and C (μmol) the amount of the target compound (x, i.e., LA or GA) or DMSO (internal standard).

The degree of hydrolysis of the polymer Y (%) is the sum of the yields of the individual hydrolysis products after multiplying by the corresponding proportions. This was calculated according to Equation (5)):(5)$$Y=\underset{x}{\sum }\frac{C_{x}f_{x}}{{\mathrm{ThC}}_{x}}\times 100$$
where C_x_ (μmol) represents the amount of monomer released, f the molar fraction of said monomer incorporated in the polymer and ThC (μmol) the theoretical amount of monomer upon complete hydrolysis.

The amount of dissolved LA or GA relative to the total amount of hydrolysed monomers (F_x_) was calculated according to Equation (6):(6)$$F_{x}=\frac{I_{x}/N_{x}}{I_{\mathrm{LA}}/3+I_{\mathrm{GA}}/2}\times 100$$

Here, I represents the ^1^H NMR peak area and N the number of protons corresponding to the integrated peak(s) (x, i.e., LA or GA).

## 3. Results and Discussion

### 3.1. Biodegradation in Soil of PLGA and Its Monomers 

The biodegradation of PLGA12/88 and PLGA6/94 at 25 °C in soil was followed over time, together with cellulose and PLA as comparisons (Figure 3a). After 53 days, 53(±9)% of PLGA6/94 and 37(±2)% of PLGA12/88 was converted into CO_2_. The PLGA6/94 therefore degraded faster than the PLGA12/88 and the cellulose, which showed comparable degradation. PLA, on the other hand, shows little biodegradation (<5%) at room temperature within the timeframe of this experiment, which is in agreement with what is known from literature [32].This indicates that increasing the GA amount in PLGA copolymers increases the degradation rate. 

The biodegradation in soil of the building blocks of PLA and PLGA (glycolic acid, lactic acid, lactide) was also studied (Figure 3b). As expected, the monomers degrade much faster than the polymers. The mineralisation rates of glycolic acid and lactic acid are even comparable to that of glucose. The observation that the polymers made up of these building blocks degrade at clearly slower rate is in line with the commonly considered theory that the hydrolysis of the ester bonds is the rate limiting step for biodegradation of polyesters in soil [15]. 

The lag phase, however, was longer for the building blocks than for the polymers (Figure 3b). The soil pH for the monomer and cyclic diester experiments increased from around 4 at the start of the experiment to around 6 after the incubation time (Table A2), which indicates that these acidic test substances were mineralised. The longer lag phase observed for the monomer biodegradation experiments could be caused by the high initial concentrations of the acidic monomers, which could inhibit biological activity, probably due to their acidity. The fact that the lactide tests showed low pH initially suggests it was already hydrolysed to lactic acid at the start of the experiment.

### 3.2. Non-Enzymatic PLGA Hydrolysis

Hydrolysis in nature can, in principle, occur via non-enzymatic and enzymatic pathways. Enzymatic hydrolysis requires specific hydrolases, which are typically present in fungi and bacteria. 

PLGA is known to be biodegradable for uses in the biomedical field, which has been reported widely both in vivo and in vitro [10]. Non-enzymatic hydrolysis of ester bonds is generally considered an important pathway for its (bio)degradation [10]. Therefore, non-enzymatic hydrolysis of PLGA12/88 and PLGA6/94 is also relevant in terms of environmental biodegradability. Several studies reported hydrolysis of PLGA with high GA content [33,34]. In order to better understand the role non-enzymatic hydrolysis plays in biodegradation of these polyester, NMR experiments with PLGA12/88 and PLGA6/94 were performed. 

Figure 4 shows representative ^1^H NMR spectra of PLGA 12/88, PLGA 6/94 and PLA hydrolysate in D_2_O with DMSO as internal standard (2.73 ppm). Singlets at 4.20 ppm represent the CH_2_ protons of glycolic acid (position 2) and peaks between 1.40–1.65 result from CH_3_ protons of lactic acid (position 1), including small peaks assigned to dimers (a’) and trimers (a’’), respectively, the integrations of which, relative to the DMSO, were used to quantify the degree of hydrolysis. 

These peaks were already observed at the start of hydrolysis, which suggests either the presence of residual monomers and oligomers or that hydrolysis had already started in the air prior to starting this experiment. Although the peaks of LA dimer and trimer were observed, the monomer peaks of GA and LA were the most significant peaks present in the NMR spectra, especially in the later phase, which indicates that PLGA will eventually convert into its monomers as the main end products (Figure A1). 

Figure 5 shows the degree of hydrolysis for PLGA6/94, PLGA12/88 and PLA calculated from the amount of hydrolysed monomers. The hydrolysis of both PLGAs is significantly faster than the hydrolysis of PLA. This is in agreement with research published by Li et al. [35]. Initially the formation of lactic acid (including dimers and trimers) and glycolic acid from PLGA6/94 is clearly faster than that of PLGA12/88. This suggests that a higher LA content in the polymer reduces the rate of hydrolysis, which makes sense based on the fact that pure PLA shows slow hydrolysis. In the literature, it is described that lactate ester groups have a higher steric hindrance than glycolate ester groups for hydrolysis, which could reduce the accessibility to water and explain this phenomenon [36]. 

Starting at around week 20, the hydrolysis rate of PLGA12/88 appears to be faster than that of PLGA6/94. Furthermore, the rate of monomer formation from PLGA6/94 starts to decrease at an earlier stage than the rate decrease for PLGA12/88. As a result, the lines intersect after 50 weeks, after which the overall monomer yield from PLGA12/88 is higher than that for PLGA6/94. It is known that PLGA12/88 is amorphous, and PLGA6/94 is semi-crystalline, which could explain this observation. It is expected that for PLGA6/94 the amorphous fraction degrades first, followed by the crystalline fraction that degrades slower, overall leading to the hydrolysis rate dropping below that of PLGA12/88 after 60 weeks. 

By comparing Figure 3 and Figure 5, it becomes clear that biodegradation of PLGAs in soil at ambient conditions is significantly faster than their hydrolysis at ambient conditions in water: 50% conversion of PLGA6/94 to CO_2_ in 7 weeks and 12–16% hydrolysis of PLGA6/94 to soluble monomers and oligomers in the same time period. From this, it can be concluded that non-enzymatic hydrolysis could play a role in PLGA soil biodegradation but that enzymatic hydrolysis is significantly faster and therefore dominant in PLGA biodegradation.

Figure 6 shows the individual yields (a) relative to their maximum theoretical yield and amounts (b) of glycolic acid and lactic acid for all polymer hydrolysis experiments at 25 °C in D_2_O. The relative yields of lactic acid are higher than those of glycolic acid in the same experiment (Figure 6a). This indicates that initially a higher amount of LA and a lower amount of GA is released than should be expected from the incorporated ratio in the polymer. This in turn suggests that GA–LA ester bonds are more susceptible to hydrolysis than GA–GA ester bonds. LA–LA ester bonds are not likely to play a major role here, given PLA’s slow hydrolysis. The overestimation of the LA yield is likely caused by the low absolute amounts relative to the total amount of monomers, which impacts the accuracy. This also explains why this is more pronounced for the PLGA6/94 samples than for the PLGA12/88 samples.

Although initially the amounts of LA were in the same range for both PLGAs, after approximately 20 weeks the amount of PLGA12/88-LA started increasing relative to that of PLGA6/94-LA (Figure 6b). After approximately 50 weeks, a similar trend is observed for glycolic acid amounts, where the amount of GA released from PLGA12/88 overtakes that of PLGA6/94 (Figure 6b). This relative rate decrease of PLGA6/94-GA could be explained by the lack of GA-LA ester bonds at that point in time. 

Figure 7 shows the percentage of released glycolic acid (GA) and lactic acid (LA, including dimers and trimers), relative to their sum (i.e., GA/(GA + LA) or LA/(GA + LA)) observed during hydrolysis of the PLGA12/88 and PLGA6/94 copolymers. This provides a visualisation of the ratio of the sum of monomers released in time. It shows that the initial monomer ratio is close to 50/50 for PLGA12/88 and close to 30/70 (LA/GA) for PLGA 6/94. This can be interpreted in two ways: either the polymer contains sections with different monomer ratios, of which the sections highest in LA content hydrolyse first, or the LA-GA bonds across the polymer are hydrolysed and released at a higher rate. This also shows that as the hydrolysis progresses, the GA content increases in the remaining polymer, which in turn could increase the degree of crystallinity and slow down its hydrolysis. The hydrolysis did slow down clearly after about 20 weeks for PLGA6/94 and after about 40 weeks for PLGA 12/88 (Figure 5) [37,38,39].

The percentage of GA slowly increased to 89% and that of LA decreased to 11%. This LA/GA ratio of 11/89 is similar to the feed ratio for synthesis, however, lower than the ratio of 6/94, which was obtained after complete hydrolysis. A GA content higher than the feed ratio could be expected due to the known higher reactivity of GA (primary alcohol) compared to LA (secondary alcohol) and the possible evaporation of the latter during synthesis at the selected reaction conditions. 

The higher initial rate of hydrolysis of PLGA6/94 compared to PLGA12/88 can be explained by the higher content of GA leading to more hydrophilic polymers, which facilitates more water uptake [40]. This could also explain the relatively slow hydrolysis of PLGA12/88 in the first several weeks. However, PLGA12/88 overtakes PLGA6/94 after around 60 weeks. This makes sense, taking into account that PLGA6/94 is a semi-crystalline copolymer and PLGA12/88 is amorphous. The amorphous areas are expected to be more accessible and therefore more reactive than the crystalline areas [41]. Alternatively, this may also attribute to higher content of LA-GA ester bonds of PLGA12/88. In short, there are two competing factors that determine the relative hydrolysis rate of PLGA copolymers with high GA content: on one hand, higher LA content leads to less hydrophilicity, mainly affecting the early stages of hydrolysis, and conversely, the presence of crystalline areas in copolymers with higher GA content seems to slow down the hydrolysis in the later stage. 

The biodegradation of PLGA in soil is much faster than the non-enzymatic hydrolytic degradation in heavy water. A higher percentage of carbon from PLGA6/94 compared to that for PLGA12/88 was converted into CO_2_ within the timeframe of the biodegradation experiments. At the same time PLGA6/94 yielded more monomer from the hydrolysis experiments than PLGA12/88 in the early stages of the hydrolysis experiments (before reaching 50% yield). Furthermore, PLA showed little conversion to CO_2_ and limited hydrolysis to lactic acid at ambient temperature. It is also clear that the monomers themselves (LA and GA) all readily degraded to CO_2_. These results indicate that enzymatic hydrolysis is the rate limiting step in the biodegradation of these PLGA polyesters.

The fact that PLGAs with high GA content hydrolyse in water at a relatively high rate, potentially translates well to marine environments. Although temperatures might not always be as high, they are not expected to remain for decades. Bagheri et al. observed the complete degradation (mass loss) of poly(D,L-lactide-co-glycolide) (PLGA50/50) plastic films in seawater within a year, compared to 0% degradation for PLA [42]. PLGA co-polymers are already known to be biocompatible and therefore used in biomedical applications, which means they likely pose less of a risk when ingested by fauna [10].

### 3.3. PLGA versus Commodity Barrier Plastics 

The combination of its biodegradability and good barrier properties makes poly(glycolic acid) (PGA) an interesting polymer. It is, however, difficult to process because of its high degree of crystallinity [9]. As a result, it is not suitable for most of the bulk packaging applications. Conversely, PLA is easier to handle, but does not have favourable barrier properties and shows little biodegradation at room temperature (Figure 3a) [43]. Combining the two in copolyesters, i.e., adding small proportion of LA to PGA, results in more processable (than PGA) polymers that still possess good barrier properties and biodegradability.

Conventional polymers used for their barrier properties in packaging are PET, polypropylene (PP) and polyethylene (PE). Negligible weight loss or biodegradation of those polymers was observed after 8 months to 2 years in soil, which is not surprising given the omnipresent plastic litter from these plastics [44,45,46]. Bio-based versions of these plastics are in various stages of development, which deals with the CO_2_ issue, but not with the plastic waste problem [47]. Using non-biodegradable plastics for short-lifetime applications, such as packaging, makes little sense when littering is almost impossible to prevent and/or composting is used for their end-of-life treatment: micro- and nanoplastics will undeniably be released and end up in the environment [48,49]. From this point of view PLGA type materials make sense as packaging materials for the future. An application that appears to make sense is that of paper coating, as the PLGA degrades even faster than the paper (cellulose) it is coated on. 

Apart from conventional polymers, polyhydroxybutyrate (PHB) and polyhydroxybutyrate*-co-*hydroxyvalerate (PHBV) have attracted interest for packaging applications since they are bio-based and biodegradable [50]. Although PHB and PHBV also have good barrier properties, PLGAs have better thermal properties (glass transition temperature (T_g_), around 40 °C vs. lower than ambient temperature), which allows for more applications [9,43,51].

### 3.4. Applicability of the Research Method 

The Respicond with 95 vessels allows parallel testing of 16 materials with five replicates, including two abiotic controls, six soil blanks and three reference vessels. Assuming 6-month incubation, it would take 8 years for the same experiments with 10 individual reactors including one abiotic control, three soil blanks and two reference vessels. The benefit of this setup is that it enables real-time, accurate, online and high-throughput measurements. The high-throughput, parallel vessel setup allows for testing the biodegradability of (new) plastic materials and for evaluating the effect of variables such as polymer molecular weight, polymer crystallinity, and the influence of environmental factors such as pH, temperature and moisture on the biodegradation rate. Especially considering the slow nature of the biodegradation of plastics, this high-throughput parallel approach with automated CO_2_ release monitoring is key for obtaining significant amounts of data within a realistic time frame. The required global shift to sustainability is expected to lead to an increase in the development of novel materials, for which fate-in-nature should be an important parameter to research. To efficiently achieve this, high-throughput testing is a must.

The NMR method used allows both qualitative and quantitative analysis for soluble hydrolysis products in a relatively quick and easy way. Compared to chromatographic methods, it is far less time-consuming. Although this method requires water-soluble products with non-overlapping identical peaks, it does not require intermittent (invasive) sampling, which requires lots of replicates, and the typically time-consuming samples pre-treatment. Especially, long-term experiments, such as the hydrolysis of polyesters at ambient temperature, will benefit from this.

For scientific purposes, one would like to determine the biodegradation rate of materials quantitatively and accurately, and understand the processes involved. From a more societal and industrial perspective it is important to have an indication of the timeframe in which a material is expected to degrade, since the precise values can be influenced by many factors that are difficult to determine exactly (different soils, seasonal shifts, etc.). It is, in a sense, much more important to gain understanding on if and how polymers will degrade in nature and to make sure that they break down to CO_2_ and water, rather than to non-degradable oligomers and small molecules, as to avoid build-up of waste. Using references such as cellulose and glucose helps to provide a baseline for how long materials will linger when not disposed of properly. 

For certain applications, it makes sense to design for degradability, for example in agriculture. For most applications, however, reuse and recycling are still preferred over disposal, for obvious reasons, and environmental fate research relates to what happens with materials when they inadvertently end up in nature. For any novel materials to be developed, the platform and methodology described in this paper could provide essential clues in researching fate-in-nature at an early stage of the development, where these issues can still be dealt with without escalating cost. Especially polyesters could be tuned for biodegradability in earlier phases of research by performing these kinds of degradability studies in parallel with materials development. This can only realistically work using a high-throughput setup, given the timeframe. A high-throughput system also provides the opportunity to study the mechanism of polymer biodegradation.

## 4. Conclusions

In this study, a parallel automated respiration platform (Respicond) was successfully used for plastic biodegradability testing on poly(lactic*-co-*glycolic acid) copolyesters high in glycolic acid content: PLGA12/88 and PLGA6/94. Their conversion to CO_2_ at ambient temperature (25 °C) in soil was monitored, providing highly reproducible data. The biodegradability of copolyesters was comparable to that of cellulose and much higher than that of PLA, as around 50% of PLGA6/94 and 40% of PLGA12/88 was converted into CO_2_ within 8 weeks. Furthermore, faster biodegradation was observed for monomers (LA, GA and lactide) than for the polymers. 

In parallel, the non-enzymatic hydrolysis of these polymers was also studied, using NMR and D_2_O. This showed that the polyesters consisted of sections with different sensitivity towards hydrolysis, leading to an uneven release of monomers. Over 60% of both PLGAs was hydrolysed within 2 years.

The PLGA copolyesters with high glycolic acid content show the potential to biodegrade to CO_2_ and biomass in a matter of months. This, together with the combination of good oxygen and moisture barrier properties reported previously, makes them interesting for food packaging. Especially considering a PLGA-coated paper packaging, the biodegradable film would result in a home compostable waste, which in areas that lack logistics for collection and recycle is an interesting alternative to incineration in the open air. 

This research also showed that a high-throughput platform allows high quality study of the biodegradability of novel polymers with various compositions in a time-efficient way, which can help scope research into novel materials already in the early stages at limited cost. This is of considerable importance, given the global plastic waste crisis, which will force a transition to materials that do not endlessly linger in nature. Given the fact that CO_2_ evolution is always calculated against the natural CO_2_ evolution in the soil, materials or chemicals that degrade slowly are more difficult to study. 

## Figures and Tables

**Figure 1 polymers-14-00015-f001:**
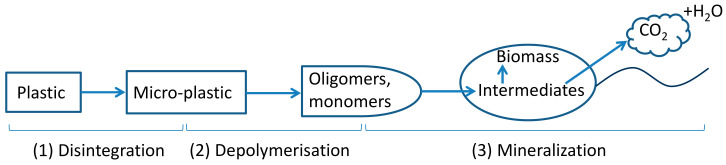
Plastic biodegradation under aerobic conditions. Arrows represent carbon flow.

**Figure 2 polymers-14-00015-f002:**
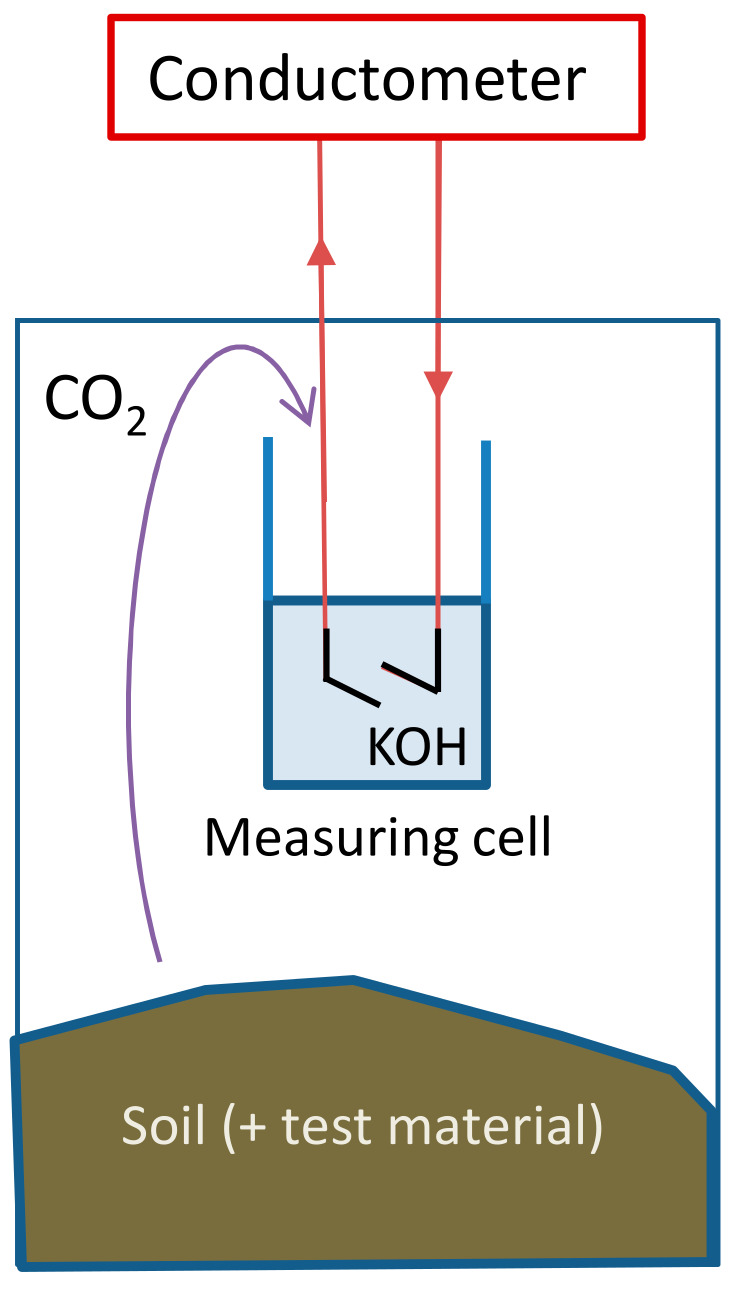
Measuring unit of the parallel respirometer.

**Figure 3 polymers-14-00015-f003:**
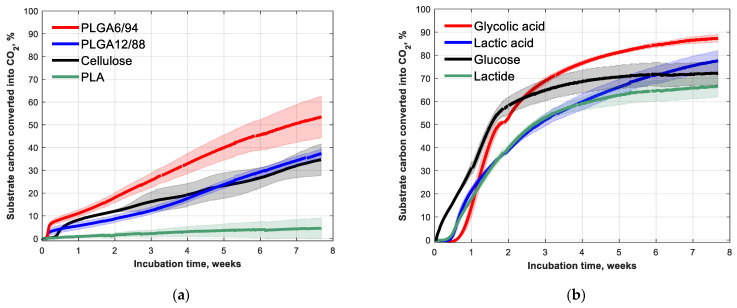
Fifty-three-day biodegradation curves of (**a**) PLA, PLGA12/88, PLGA6/94 and cellulose (references) and (**b**) glycolic acid, lactic acid, lactide and glucose (building blocks) with approximately 5 mg (substrate) carbon g^−1^ dry soil at 25 °C. Mean biodegradation (lines) were plotted. The shaded area represents the standard deviation (calculated per point) of at least three replicates, except for glucose, in which case it represents the range of the duplicates.

**Figure 4 polymers-14-00015-f004:**
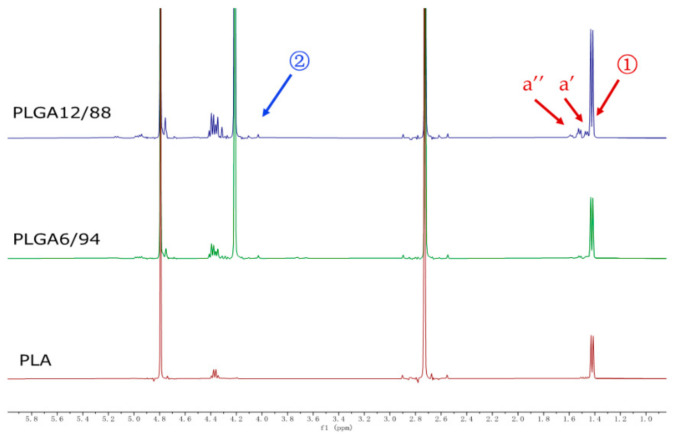
^1^H NMR spectra of PLGA 12/88, PLGA 6/94, and PLA hydrolysis in D_2_O with 2.0 mg dimethyl sulfoxide as internal standard in 45 weeks.

**Figure 5 polymers-14-00015-f005:**
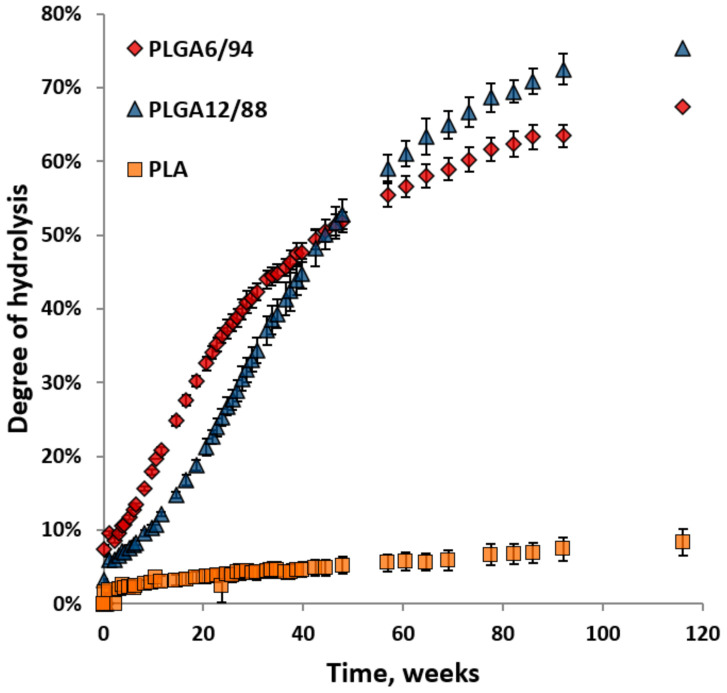
Degree of hydrolysis for PLGA6/94, PLGA12/88 and PLA versus time over 116 weeks at 25 °C in D_2_O. The points represent the averages of triplicate experiments, with the error bars representing the standard deviation.

**Figure 6 polymers-14-00015-f006:**
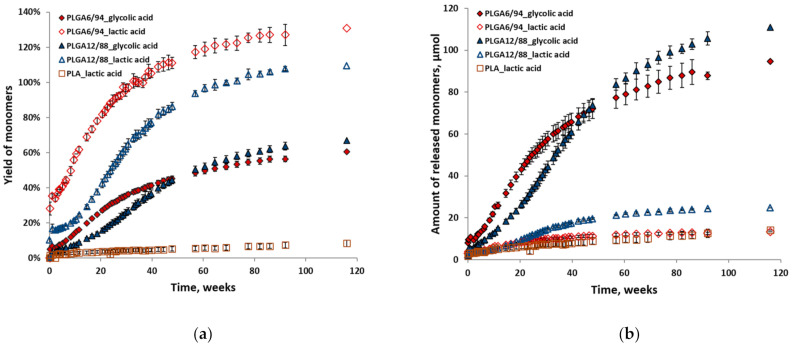
Individual yields (**a**) and amounts (**b**) of monomers (glycolic acid and lactic acid) versus time from hydrolysis of PLGA6/94, PLGA12/88 and PLA at 25 °C in D_2_O. The points represent the averages of triplicate experiments, with the error bars representing the standard deviation.

**Figure 7 polymers-14-00015-f007:**
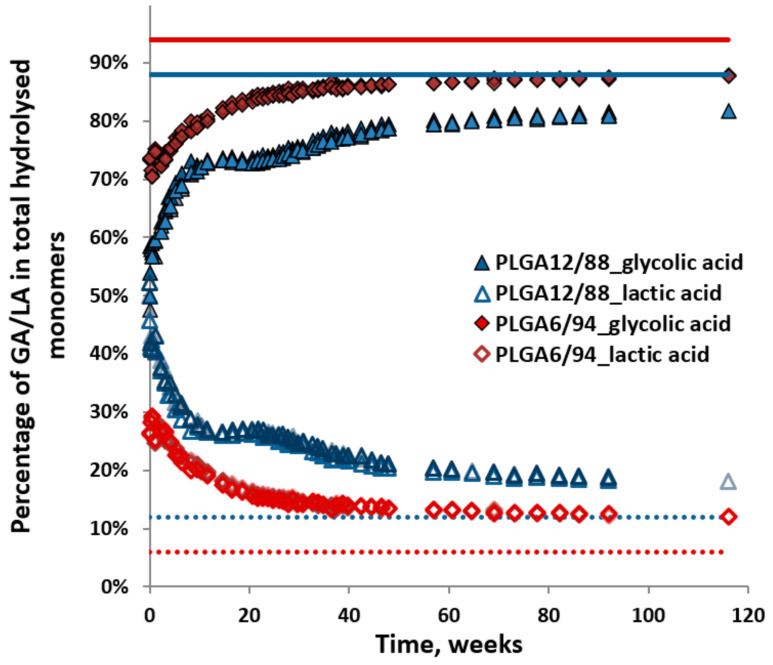
Percentages of dissolved glycolic acid and lactic acid relative to the total amount of hydrolysed monomers in time for PLGA12/88 and PLGA6/94. Triplicates were plotted. Dotted and solid lines show the starting composition of the polymer.

**Table 1 polymers-14-00015-t001:** Properties of soil.

Properties	Values
Sand/Silt/Clay (%)	90/5/1 *
Organic carbon (mg g^−1^)	18.29 *
Nitrogen (mg g^−1^)	0.97 *
C:N (g C g^−1^ N)	22 *
Phosphate (µg g^−1^)	4.6 *
pH (0.01 M CaCl_2_)	5.9
Cation exchange capacity (mmol^+^ kg^−1^)	60 *
Field capacity (g water 100 g^−1^ dry soil)	33.3

* Values taken form Schlemper et al. (2017) [27].

**Table 2 polymers-14-00015-t002:** Thermal transitions for PLGA copolymers and PLA recorded from DSC (10 °C min^−1^). Polycondensation (PC), ring opening polymerisation (ROP).

Polymers	T_g_ (°C)	T_m_ (°C)	Structure
PLGA12/88 (PC)	40	-	Amorphous
PLGA6/94 (PC)	31	186	Semi-crystalline
PLA (ROP)	45	165	Semi-crystalline

## Data Availability

The data presented in this study are available on request from the corresponding author.

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
