# Peer review of "Biodegradation and Non-Enzymatic Hydrolysis of Poly(Lactic-co-Glycolic Acid) (PLGA12/88 and PLGA6/94)"

_polymers, 2021, doi:10.3390/polym14010015_

Round 1

Reviewer 1 Report

1. Figure 1 is not cited. Please cite it and explain it. 2. Rewrite the conclusion by adding the limitations and future scope of this work.

Author Response

We would like to thank reviewer 1 for his/her time to review our paper and for the valuable feedback.

1. Figure 1 is not cited. Please cite it and explain it.

We apologize for the unclarity. Figure 1 was cited and explaned in lines 83-93 but the Figure has been moved up to Line 94. It is now directly preceding the paragraph citing and explaining it in the revised version.

2. Rewrite the conclusion by adding the limitations and future scope of this work.

We thank reviewer 1 for this suggestion. We have revised the conclusions to now discuss future scope for the polymer in lines 501-514 in the revised version and limitations of the method in lines 512-514 of the conclusion.

Reviewer 2 Report

This paper reports on the “Biodegradation and non-enzymatic hydrolysis of poly(lactic-co-glycolic acid) (PLGA12/88 and PLGA6/94)”. Introduction and conclusion, methodology and reference, results and discussion seems be corrected.

Manuscript may be published in Polymers.

Author Response

we thank reviewer 2 for his/her time to review our paper and for the valuable feedback.

This manuscript is a resubmission of an earlier submission. The following is a list of the peer review reports and author responses from that submission.

Round 1

Reviewer 1 Report

The manuscript describes the attempt to use an automatized equipment in the polymer biodegradation measurements of poly(glycolic), poly(lactic acid) and two copolymers of them, also the results were compared with micro-cellulose.

Several sentences in the manuscript needs the inclusion of the appropriate bibliographic references for example in page 2 lines 66-67, in the same page lines 68-69, and lines 77-79, page 3 line 111-115.

Some experimental details are missing in page 4 line 135 A value, in table 1 the properties values of soil were determined or were taken from the literature? I not clear the amount of soil added to each experiment, for example, according to the text in line 158, the authors that 15 g of soil mixture was added and in table 1 the soil contain 18.29 mg/g of carbon and at least contain  274.35 mg of carbon and in page 5 line164, the authors mention that the typical amount of organic carbon introduced to the test tube  to the dry soil was kept at around 5 mg C/g dry soil, the entire description  needs to be improved.

The methodology for the preparation of the copolymers is not given, the physicochemical properties for all material are missing, for example the molecular weight, type of copolymers, elemental analysis, crystallinity, type of enantiomer, the particle size of polymer sample, among others.

In my personal opinion page 5 line 186, the ThCO2 amount needs to be the determined carbon % w/w from the elemental analysis of each material.

In the results section, why the authors do not consider the PGLA polymer in the degradation studies?

The authors consider L(+) Lactc acid, and D(+) Glucose has some influence in the degradation the type of isomer? And lactide isomer? Any characteristic were given.

In page 7 line 229 the authors mention the standard deviation, also in the line232 mention that the reproducibility of data is very good, I not agree with this affirmation, for example in figure 3a at 30 days of degradation the standard deviation for PLGA the results are at least 5 and increases to the double at 50 days. Cellulose at least the value around 10. In my personal opinion the results requires more discussion for example why glucose and cellulose have the highest standard deviation? According to the molecular weight and crystallinity why the cellulose have bigger biodegradation compared to the PLA for me, make no sense.

On my personal opinion, the manuscript needs more experimental data in order to postulate more solid conclusions

Reviewer 2 Report

This work reports on the applicability of a respirometer to the study of the biodegradation of poly(lactic-co-glycolic acid) copolymers in comparison with cellulose.

Although the simplicity of the technique for this purpose is an interesting information to take into account for polymers biodegradation studies, it does not fullfil the standards to be published as a research article in the present form due to the following reasons:

  • It is not clear the goal of the investigation. I am not sure if the authors aim to study the biodegradation of the polymers or to study the suitability of the technique. Nevertheless in both cases the selected variables are not enough and experimental results are not representative. I mean, if polymers biodegradation is explored, more polymers or specific variables from the same polymers (molecular weight, branched/linear..) should have been considered. In case, the focus of the sutudy is the technique, results must have been compared with other techniques, and accuracies and sensibilities must have been also specified.
  • The experimental results and the theoretical information are not equilibrated. Experimental results are limited to Figure 3.a) and discussion to lines 231-244, because figure 3. B) (monomers degradation) does not provide further information when polymers degradation is explored. These results are not enough to complete a research paper.
  • Other`s results and those from this paper are mixed in the discussion to some extent that lines 247-268 are close to be a minireview, or even an opinion article (line 291) unconnected with the obtained results. If PLA, PHB, PET, PP..polymers are also interesting and comparable, authors should have also tested these polymers with this technique.
  • Results and discussion section cannot start with a figure.
  • 3.1 Section does not correspond with results. It must be moved to the introduction.
  • Typographical errors in line 231.

Reviewer 3 Report

The manuscript entitled “Automated parallel biodegradation testing of poly(lactic-co-glycolic acid) (PLGA20/80 and PLGA10/90)” by Yue Wang et al. reports an efficient system Respicond parallel respirometer for studying polymer (bio)degradation of poly(lactic-co-glycolic acid) polyesters and their monomers. The method is based on quantifying the process of polymer biodegradation by CO2 evolution during mineralization under aerobic conditions, thus not specific to a material characteristic and can be applied to a variety of biodegradable polymers as well as their building blocks. This study tries to deal with an important subject related to bio-based polymers. However, the amount of data included in this manuscript is quite less. The results and discussion part of this article rather includes general writing than providing an in-depth explanation of the obtained results. As claimed by the Authors in lines 215-220, page 6, the Reviewer wants to see more data on polymer biodegradation by changing variables such as polymer molecular weight, polymer crystallinity, and the influence of environmental factors like pH, temperature, and moisture on the biodegradation rate. Considering the concern of the Reviewer about the lack of data, the Reviewer recommends this manuscript to be submitted again after judicious inclusion of large and meaningful data.